# Sodium-Calcium Exchanger 2: A Pivotal Role in Oxaliplatin Induced Peripheral Neurotoxicity and Axonal Damage?

**DOI:** 10.3390/ijms231710063

**Published:** 2022-09-02

**Authors:** Elisa Ballarini, Alessio Malacrida, Virginia Rodriguez-Menendez, Eleonora Pozzi, Annalisa Canta, Alessia Chiorazzi, Laura Monza, Sara Semperboni, Cristina Meregalli, Valentina Alda Carozzi, Maryamsadat Hashemi, Gabriella Nicolini, Arianna Scuteri, Stephen N. Housley, Guido Cavaletti, Paola Alberti

**Affiliations:** 1School of Medicine and Surgery, University of Milano-Bicocca, 20126 Milan, Italy; 2NeuroMI (Milan Center for Neuroscience), 20126 Milan, Italy; 3Integrated Cancer Research Center, School of Biological Sciences, Georgia Institute of Technology, Atlanta, GA 30332, USA

**Keywords:** NCX2, voltage-operated ion channels, chemotherapy-induced peripheral neurotoxicity, chemotherapy induced peripheral neuropathy, axonal damage, axonal hyperexcitability, immunofluorescence, immunohistochemistry, nerve excitability testing, neuropathology, neuroprotection

## Abstract

Oxaliplatin (OHP)-induced peripheral neurotoxicity (OIPN) is a frequent adverse event of colorectal cancer treatment. OIPN encompasses a chronic and an acute syndrome. The latter consists of transient axonal hyperexcitability, due to unbalance in Na^+^ voltage-operated channels (Na^+^VOC). This leads to sustained depolarisation which can activate the reverse mode of the Na^+^/Ca^2+^ exchanger 2 (NCX2), resulting in toxic Ca^2+^ accumulation and axonal damage (ADa). We explored the role of NCX2 in in vitro and in vivo settings. Embryonic rat Dorsal Root Ganglia (DRG) organotypic cultures treated with SEA0400 (SEA), a NCX inhibitor, were used to assess neuroprotection in a *proof-of-concept* and *pilot* study to exploit NCX modulation to prevent ADa. In vivo, OHP treated mice (7 mg/Kg, i.v., once a week for 8 weeks) were compared with a vehicle-treated group (*n* = 12 each). Neurophysiological and behavioural testing were performed to characterise acute and chronic OIPN, and morphological analyses were performed to detect ADa. Immunohistochemistry, immunofluorescence, and western blotting (WB) analyses were also performed to demonstrate changes in NCX2 immunoreactivity and protein expression. In vitro, NCX inhibition was matched by ADa mitigation. In the in vivo part, after verifyingboth acute and chronic OIPN had ensued, we confirmed via immunohistochemistry, immunofluorescence, and WB that a significant NCX2 alteration had ensued in the OHP group. Our data suggest NCX2 involvement in ADa development, paving the way to a new line of research to prevent OIPN.

## 1. Introduction

Oxaliplatin (OHP)-induced peripheral neurotoxicity (OIPN) is a frequent toxicity, experienced by a growing population of colorectal cancer survivors that can be long-lasting, or even permanent [1,2], altering patients’ quality of life [3,4]. At the state of the art, there is no efficacious preventive or curative treatment for OIPN [5]. One of the reasons for this unmet clinical need is the incomplete knowledge on axonal damage (ADa) mechanisms [6,7]. Therefore, robust experimental evidence is still required to devise novel treatments. OIPN is characterised by two different conditions. A *chronic sensory neuropathy* characterised by ADa, known as chemotherapy-induced peripheral neurotoxicity (CIPN), which is one of the commonest late toxicities of several anticancer drugs (platinum drugs, taxanes, vinca alkaloids, proteasome inhibitors, epothilones, and thalidomide [8,9,10,11]). OIPN, however, is also characterised by a *specific acute neurotoxicity syndrome*, as soon as after the first chemotherapy cycle. Acute signs/symptoms occur nearly in all OHP-treated patients mirroring an axonal hyperexcitability state: transient cold-induced paraesthesia at limb extremities, cold-induced dysesthesia at oral cavity/pharynx, jaw spasm, and cramps, lasting mainly the 24–72 h after each OHP administration [12,13]. A possible causative link between acute and chronic OIPN was hypothesised. Even though acute OIPN is transient and never dose-limiting, the underlying axonal hyperexcitability could be potentially linked to cellular stress and, therefore, ADa [14,15]. Several preclinical in vitro observations showed sodium voltage-operated ion channel (Na^+^VOC) alterations are involved in acute OIPN. In the literature, alternative mechanisms involving also other channels were hypothesised [16], but there are robust in vitro data, in vivo data, and in silico modelling demonstrating that Na^+^VOC alterations are pivotal and other channels/transporters, if involved, are a secondary element [17,18,19]. We already provided clinical observations in line with this: a more severe chronic OIPN was associated with a more pronounced acute OIPN and specific Na^+^VOC polymorphisms [20,21], whereas K^+^ voltage-operated ion channels ones were not associated with OIPN [22]. Exploiting nerve excitability testing (NET), Na^+^VOC dysfunctions were also demonstrated in OHP-treated patients, before ADa ensued [15,23,24]. On the basis of these observations, we tested topiramate (TPM), a known Na^+^VOC modulator approved for clinical use in a refined rat model [25]; complete neuroprotection was demonstrated via a refined set of multiple outcome measures [26]. However, the exact causative link between acute Na^+^VOC dysfunctions and chronic ADa is still to be investigated. Na^+^VOC alterations are only transient [1,18] and functional, thus OIPN is quite different respect to the condition of “sick-NaV channels” extensively described by Morris et al. [27], in relation to traumatic/ischemic/inherited conditions; in these cases, structural degradation of axolemma bilayer is present. Therefore, we hypothesised that OHP-related ADa was due to a downstream event respect to **Na^+^VOC impaired functioning**, which would be a transient, functional, **upstream event**. The Na^+^/Ca^2+^ exchanger (NCX) family could play a pivotal role in this regard, in particular its isoform 2 (NCX2); NCX2 is widely expressed along distal parts of the axons together with Na^+^VOC 1.6, 1.8, and 1.9 [28], as well as in dorsal root ganglia (DRG) [28], the primary target of OHP neurotoxicity [29]. NCX is a bidirectional transporter dependent on the gradient of the two involved ions, Na^+^ and Ca^2+^ [30]. NCX is one of the key regulators of Ca^2+^ homeostasis and, in normal conditions, NCX extrudes Ca^2+^ from cells (*forward mode*), but, in case of ion unbalance (e.g., enhanced Na^+^ influx, which determines an aberrant neuronal depolarization [31]) NCX starts to operate in the *reverse mode*, resulting in Ca^2+^ importing [32]. Ca^2+^ overload is a relevant contributor leading to cellular damage due to activation of Ca-sensitive calpain, phospholipases, and nitric oxide synthase [28,33,34]. Ca^2+^ intraneuronal levels should be tightly controlled; Ca^2+^, in fact, is one the major triggers of neurotransmitter release [35] and is essential for several other functions related to neuronal excitability [36], integration of signals [36], synaptic plasticity [37], gene expression [38], metabolism [39], and programmed cell death [40]. Therefore, an uncontrolled level of this ion can lead to neuronal damage and death [41]. Since NCX *reverse mode* activation leads to alterations in Ca^2+^ balance, it is an intriguing **possible downstream event** leading to ADa. Preliminary in vitro or ex vivo findings suggested that NCX2 has a relevant role in ADa in peripheral nerves [34]. Earlier studies were done on rat optic nerves (that strictly anatomically and embryologically speaking are part of the central and not the peripheral nervous system), showing that metabolic anoxia-induced stress resulted in a persistent Na^+^ influx via Na^+^VOC, and this was also related to the metabolic impairment of the Na^+^/potassium ATP-ase pump, leading to NCX *reverse mode* activation and subsequent Ca^2+^-related toxicity in axons [42]; similar findings were then replicated in peripheral nervous system myelinated axons exposed to anoxia [43,44]. Ca^2+^ toxicity due to the *reverse mode* activation was related to several neurological diseases, and the possible beneficial role of its modulation has been described in different conditions such as brain ischemia [31], Alzheimer disease [45], and amyotrophic lateral sclerosis [46]. However, further observations are still required. Therefore, we performed in vitro and in vivo experiments to shed light into the potential role of NCX2 in ADa development. In the in vivo study, we provide evidence of potential NCX2 involvement in OHP-related ADa, and in the in vitro part we propose a *proof-of-concept* and *pilot* study of NCX modulation effects.

## 2. Results

### 2.1. In Vitro Observations

The role of NCX2 in OIPN was investigated in vitro using the well-established experimental DRG explant model obtained from embryonic Sprague Dawley rats; neurite elongation is the key parameter to assess neurotoxicity and neuroprotection [47]. As shown in Figure 1, DRG exposed to culture medium emitted long neurites, which shortened after exposure to OHP for 24 h (−20%, *p*-value < 0.05 vs. control) and 48 h (−60%, *p*-value < 0.001 vs. control). DRG pre-treated with SEA0400 (20 µM), a strong and selective NCX family inhibitor [48], for 3 h did not affect neurite elongation under control conditions, and protected neurite outgrowth of DRG exposed to OHP for 24 h (+40%, *p*-value < 0.01 vs. OHP) and 48 h (+20%, *p*-value < 0.05 vs. OHP).

### 2.2. In Vivo Observations

Study design for the characterization of an in vivo comprehensive OIPN mouse model was as follows. Nerve conduction studies (NCS) and behavioural test parameters were used to verify homogeneity between the experimental groups before 1st chemotherapy administration (no statistically significant difference for all parameters). Acute OPIN was verified after 1st OHP injection monitoring NET changes in the subsequent 72 h. Chronic OIPN and, therefore, ADa development was verified via NCS, neuropathology (caudal nerve morphometry/morphology and intraepidermal nerve fiber density [IENFD]), and behavioural tests after 8 weeks of treatment. Immunohistochemistry, immunofluorescence, and western blotting were used to assess if OIPN was associated with alterations in immunoreactivity and protein expression in DRG of animals treated with OHP for 8 weeks.

#### 2.2.1. NET after the 1st Administration

NET verified acute OIPN ensued after the 1st chemotherapy cycle. Over 72 h of monitoring, the most notable change was related to an upward shift in the recovery cycle curve (*p*-value < 0.01 at 2 ms and at 2.5 ms refractoriness assessment, Mann–Whitney U-test). These findings are consistent with our previous study and suggest the onset of OHP-related axonal hyperexcitability that alters Na^+^VOC functioning, leading to an aberrant depolarization [25]. Figure 2 gives an overview of the monitoring over 72 h showing statistically significant parameters. Individual traces of each recording are provided in Appendix A.

#### 2.2.2. NCS and Behavioral Tests at the End of Treatment

A pattern compatible with polyneuropathy was observed via NCS: significant decrease in both caudal and digital nerve sensory action potential (SAP) amplitude and digital nerve conduction velocity was observed in OHP-treated animals. These findings demonstrated that relevant OHP-related ADa ensued, able to cause a functional impairment of large myelinated fibers, which are the ones tested with NCS [49]. Similarly, mechanical allodynia (Dynamic test) had ensued in OHP-treated animals (*p* < 0.001), mirroring a functional impairment ensued in small fibers (that convey painful information) [26]. Data are summarised in Figure 3.

#### 2.2.3. Caudal Nerve Morphology and Morphometry

Morphological examination of caudal nerves, harvested after sacrifice at the end of 8 weeks of treatment, demonstrated mild ADa in large myelinated fibers, matching observations obtained via NCS. Morphometrical analyses showed a statistically significant decrease in the fiber density of large myelinated fibers in the OHP treated group. Figure 4 shows representative photographs of caudal nerve morphology and statistical tests (Mann–Whitney U-test).

#### 2.2.4. Intraepidermal Nerve Fiber Density (IENFD)

Analysis of IENFD allows to formally count small nerve fibers (and eventually assess their loss), relying on cutaneous terminals. At the end of treatment, IENFD demonstrated a statistically significant decrease in treated animals compared to controls, mirroring the functional impairment that was observed at Dynamic test for mechanical allodynia. Mann–Whitney U-test significance and representative images are shown in Figure 5.

#### 2.2.5. Immunohistochemistry (IHC), Immunofluorescence (IF), and Western Blotting (WB) for NCX2

DRG harvested at sacrifice after 8 weeks of treatment were used to assess whether a difference can be found between control and OHP animals. We conducted analyses with three techniques to verify the robustness of our observations. IHC and IF showed a similar pattern of mainly cytosolic NCX2 immunoreactivity (shown in red in IF images and in dark brown in DAB-stained IHC images); a statistically significant reduction in the OHP group with respect to control was observed with both techniques, exploiting a quantitative measurement of NCX2 immunoreactivity (Figure 1C,D, respectively), and a similar reduction was demonstrated via WB analysis on the DRG pool (Figure 6E,F). This is an indirect confirmation that NCX2 *reverse mode* was activated. If *reverse mode* was activated due to an aberrant depolarization, as already demonstrated by Boscia et al. [50], a repetitive spreading of depolarisation current resulted in a downregulation of NCX2 in neurons (this is an endogenous autoprotective mechanism to prevent Ca^2+^ overload).

## 3. Discussion

So far, no mouse model has been described reproducing both acute and chronic OIPN; therefore, we first validated the robustness of the proposed model. Recently, Makker et al., in fact, characterised a mouse model showing only acute OIPN features, exploiting NET [18]. As performed in a previous study from our group in a rat model [25], we relied on a multimodal approach to demonstrated both acute and chronic OIPN had ensued [6,26]. Recovery cycle parameters at NET after the first cycle showed the typical acute OIPN pattern: OHP group showed an upward shift of the recovery cycle curve, compatible with altered Na^+^VOC functioning as already demonstrated [15,18,25]. We relied on multiple outcome measures to verify the full expression of chronic OIPN (i.e., ADa) at the end of treatment [26]. NCS were used to demonstrate the presence of a sensory axonal neuropathy; consistent with clinical observations, we observed mainly a significant drop in SAP amplitude compatible with sensory ADa [20,51,52]. Morphology of caudal nerves matched neurophysiological results, and morphometry demonstrated axonal loss of myelinated fibers. Larger diameter fibers were the ones with the most evident damage, matching clinical experience; patients mainly show sensory large fibers impairment, up to a disabling sensory ataxia [1,11,53]. Even if large fibers are the main target of OHP ADa, impairment in small fibers was clearly demonstrated in animal models [25,54] and was also observed in a small clinical cohort [55]; our model also reproduced this feature as demonstrated via IENFD reduction. Finally, allodynia was tested to demonstrate a functional impairment in small fibers; mechanical allodynia was tested, instead of temperature-triggered allodynia, to avoid possible overlap between acute OIPN and ADa, given the former is cold triggered [1,11]. Thus, the full OIPN spectrum was reproduced in our animals allowing us to perform NCX2-related observations in a setting mirroring the clinical condition.

We verified NCX2 immunolocalisation in DRG via immunohistochemistry/immunofluorescence. NCX2, quite notably, is known for being well represented in DRG [33], which are the first target in case of ADa due to OHP [11]. Even if indirectly, our data were compatible with NCX2 *reverse mode* activation. First of all, we demonstrated that the prerequisite of *reverse mode* activation ensued (i.e., Na^+^ dysfunction) via NET; this is the trigger to switch the mode of NCX functioning. If *reverse mode* was activated via an aberrant depolarization, neurons downregulated NCX to avoid a Ca^2+^ overload [50], which is the final effect of a prolonged reverse mode activation. Therefore, the 2 months of treatment and repetitive oxaliplatin administration (each injection is a trigger for reverse mode activation) were able to determine a *reverse mode* activation relevant enough that the downregulation mechanism was established; this is an endogenous and autoprotective mechanism that it is not ultimately strong enough to prevent axonal damage though, as evidenced by ADa development in OHP animals as seen at NCS, Dynamic test for allodynia, and histopathological observations of large and small nerve fibers as discussed above.

Preliminary, mainly in vitro findings suggested that NCX2 modulation might mitigate toxic neuropathies [56,57], pointing out that a pharmacological modulation of this axis can be more efficacious than the endogenous mechanisms. To further investigate this, we conducted a *proof-of-concept* and *pilot* in vitro study to test beneficial effects of NCX modulation. Among possible inhibitors, we selected SEA0400 since it is the most potent and selective NCX inhibitor [58]. Other options, such as the less strong NCX inhibitor KB-R7943, showed a rather low specificity for NCX family even at low dosages, with larger impact on off-targets with respect to SEA0400 [58,59,60]. Moreover, SEA0400 was already used in studies that investigated nervous system injury due to NCX2 *reverse mode* activation/Ca^2+^ toxicity, caused by alterations of Na^+^ currents: Matsuda et al. [58] showed that SEA0400 attenuated, dose-dependently, damage in models of brain ischemia. Koyama et al. [61] also obtained promising neuroprotection with SEA0400 in a model of brain ischemia. Notably, in a different model of peripheral neuropathy, Yilmaz et al. showed that some neuronal populations of the peripheral nervous system can be resistant to the effect of KB-R7943, but responsive to SEA0400 [57]. Therefore, SE0400 profile was the most appropriate drug for a *proof-of-concept* and *pilot* experiment to test our hypothesis. However, the specific model of disease and system we are dealing with was carefully considered to select the range of dosages to be tested. Literature data reported above, concerning the central nervous system, mostly used nM concentrations [62], but we also considered other published works to better mirror the peripheral nervous system setting; to obtain effects on axons (squid giant axon), a higher—despite still selective—dosage was require in the range of µM [63]. Moreover, in the work by Yilmaz et al., in a different model of peripheral neuropathy [57], SEA0400 was used in a microM concentrations, as well as in neuroprotection experiments relying on SH-SY5Y cells [64]. We also carefully evaluated the most appropriate OHP dose to be used, relying on the extensive review of Calls et al. [65], which presented an overview of the in vitro and in vivo models of OIPN; starting from these, we selected a dosage that was similar to the mean plasma levels in patients treated with OHP [66,67] and able to induce neurite alterations.

To ensure SEA0400 was active against down-stream events, secondary to Na^+^VOC alteration, a pre-treatment exposure to SEA0400 was performed (3 h); it was necessary, in fact, to administer SEA0400 before OHP exposure to ensure the block of NCX activity before the *reverse mode* was activated, as a consequence of Na^+^VOC unbalance, similarly to what we previously performed in rats with TPM [25]. Exploiting a consolidated approach to assess neurotoxicity in vitro [68], we observed promising data suggesting the neuroprotective role of NCX inhibition against OHP detrimental effects in neurons. This confirms, indirectly, that NCX2 *reverse mode* activation plays a role in ADa. However, some limitations should be acknowledged. Ours is a *proof-of-concept* and *pilot* study that would require further investigation. Despite the fact that our data are promising, of course, it could be argued that a combined administration of Na^+^ current and Ca^2+^ current distinct modulators might be effective too in preventing calcium overload as described in models of cardiac ischemia-reperfusion injury and cardiac glycoside toxicity [69]. However, targeting NCX with novel drugs/approaches, instead of these ion channels, is nevertheless a promising strategy since, differently from these ion channels, it is a relatively simpler family; Na^+^ and Ca^2+^ ion channels are characterised by several subtypes with relevant functional differences, which makes off-target effects quite easier [70,71]. Of course another point to be considered is that SEA0400 too, despite its specificity, could exert eventual off-target actions, in particular at higher dosages [60]; but, as stated above, the dosages we selected are expected to be quite selective in the peripheral nervous system. Overall, it can be suggested that our data pointed out that NCX2 is potentially a pivotal element in ADa and, therefore, a druggable target to be further investigated. This paves the way to a new line of research to further explore the intriguing possibility that NCX2 modulation might prevent ADa, relying on highly selective approaches to avoid off-target effects (e.g., NCX2 modulation via siRNA in in vivo OIPN models).

## 4. Materials and Methods

### 4.1. In Vitro Experiments

#### 4.1.1. Drugs

OHP was purchased from Sigma Chemical Co. (Milan, Italy). It was dissolved in physiological solution to make a stock solution of 1 mg/mL, which was diluted with medium to obtain the working concentration of 7.5 µM. This concentration has been selected in a similar range to the mean plasma concentrations in patients treated with OHP [64,70] and able to induce a relevant reduction in neurite elongation after exposure. The strong and selective NCX inhibitor SEA0400 [58] was purchased from Sigma Chemical Co. (Milan, Italy). It was dissolved in dimethyl sulfoxide (DMSO) to make a stock solution of 1 mM, which was diluted with medium to obtain the 20 µM concentration; this concentration was selected after *proof-of-concept* experiments testing a scaling dose of SEA0400 (1, 10, 20, and 30 µM) and selecting the most promising dosage in terms of neuroprotection. The inhibition effect of SEA0400 was evaluated by a 3 h pre-treatment of SEA0400 then followed by the OHP exposure. Pre-treatment was mandatory on the basis of our previously published data, demonstrating that neuroprotection against OIPN is obtained modulating ion channels/currents before OHP exposure [25].

#### 4.1.2. Dorsal Root Ganglia (DRG) Explants

DRG from 15-day-old embryonic Sprague Dawley rats were aseptically removed and cultured onto a single layer of rat-tail collagen surfaces in 35 mm dishes (4 ganglia/dish). DRG were incubated in media (MEM plus 10% calf bovine serum, 50 µg/mL ascorbic acid, 1.4 mM L-glutamine, 0.6% glucose) with 5 ng/mL NGF for 2 h in a 5% CO_2_ humidified incubator at 37 °C. DRG were pre-treated with SEA0400 (20 µM) and then also exposed to OHP 7.5 µM for 24 h and 48 h. Phase contrast micrographs of all DRG were made after 24 h and 48 h of OHP exposure. For each DRG, the longest neurite was measured. These magnified measurements were compared with a calibration grating photographed under identical conditions. Each experiment was performed three times to validate the results.

### 4.2. In Vivo Observations

#### 4.2.1. Animal and Housing

Experiments were performed on male balb/c mice weighing 18–20 g on arrival (Envigo, Bresso, Italy). Care and husbandry of animals were in conformity with the institutional guidelines in compliance with national (D.L. n. 26/2014) and international laws and policies (EEC Council Directive 86/609, OJ L 358, 1, Dec. 12, 1987; Guide for the Care and Use of Laboratory Animals, U.S. National Research Council, 1996; be carried out in accordance with the U.K. Animals (Scientific Procedures) Act, 1986 and associated guidelines, EU Directive 2010/63/EU for animal experiments, or the National Institutes of Health guide for the care and use of Laboratory animals (NIH Publications No. 8023, revised 1978). Animals were maintained under standard animal housing conditions, thus with a 12 h light–dark cycle and a room temperature and relative humidity at 20 ± 2 °C and 55 ± 10%, respectively. Drug- and vehicle-treated mice were housed separately with free access to water and food.

#### 4.2.2. Study Design

Animals were divided into 2 groups (*n* = 12 each) and treated as follows: control (CTRL) group was composed of vehicle-treated animals (5% glucose solution); OHP group was treated with OHP 7 mg/kg i.v., once a week per 8 weeks (1 qw 8 ws). Sample size for each experiment was calculated on the basis of nerve conduction velocity (NCV) reference values of our laboratory [72], assuming that the relevant difference between CTRL and OHP groups is 5 m/s (standard deviation = 7); thus, if a 2-sided 5% alpha and a 80% power is set, the sample size is 7 animals/group (www.dssresearch.com/KnowledgeCenter/toolkitcalculators/samplesizecalculators.aspx (accessed on 12 December 2019)). In each experiment the sample size was increased above the defined number (12 animals/group) in order to have enough animals to be tested at each time point, taking into account the different duration of treatment between experiments and avoiding underpowered statistical analysis in case of animal loss due to treatment. To ensure groups homogeneity, animals were randomised based on nerve conduction studies (NCS) values at baseline. Dynamic test and NCS were performed at baseline to ensure homogeneity among groups and then repeated and the end of treatment to ensure neuropathy induction on all animals/group. Nerve excitability testing was performed at 24, 48, 72 h after the 1st OHP administration to monitor changes in axonal excitability on all animals/group. At the end of the observational period (after the execution of NCS and behavioural tests at the end of treatment) animals were sacrificed and specimen for histological analysis and western blotting were harvested (3 animals/group).

#### 4.2.3. Drug Administration

OHP (Oxaliplatin 5 mg/mL solution, Accord Healthcare Limited, Durham, NC, USA), 7 mg/kg or the vehicle solution (5% glucose solution) was administered i.v. in the tail vein once a week for a period of 8 weeks.

#### 4.2.4. Dynamic Test

The mechanical nociceptive threshold was assessed using a Dynamic Aesthesiometer Test (model 37450, Ugo Basile Biological Instruments, Comerio, Italy), which generated a linearly increasing mechanical force. At each time point, after the acclimatization period, a pointed metallic filament (0.5-mm diameter) was applied to the plantar surface of the hind paw, which exerted a progressively increasing punctuate pressure, reaching up to 15 g within 15 s. The pressure evoking a clear voluntary hind-paw withdrawal response was recorded automatically and taken as representing the mechanical nociceptive threshold index. Results represented the maximal pressure (expressed in grams) tolerated by animals. There was an upper limit cut-off of 15 s, after which the mechanical stimulus was automatically terminated.

#### 4.2.5. Caudal Nerve Morphology and Morphometry

Caudal nerves were isolated and dissected out to avoid stretching. Nerves were immediately fixed by immersion in 3% glutaraldehyde in 0.12 M phosphate buffer solutions pH 7.4, post-fixed in OsO4 and embedded in epoxy resin. Then, 1.5-μm semithin sections were obtained and stained with toluidine blue. Morphometric analysis was performed on a 60x stitched image of a single nerve section per animal (3 animals/group) with Nexcope NE 920 light microscope (TIEsseLab S.r.l., Milan, Italy). The images were acquired in stitching mode using Capture V2.0 Software (Revolutionary Computational Imaging Software, TIEsseLab S.r.l., Milan, Italy) and processed by an automatic image analyser (Image-Pro Plus Software, Immagini e Computer SNC, Milan, Italy). The fiber diameter and the frequency distribution of myelinated fibers were determined, and data were analysed with GraphPad Prism Software (GraphPad Software, San Diego, CA, USA).

#### 4.2.6. IENFD

Hind paws were collected after sacrifice, and 3-mm round shapes were taken and immediately fixed in PLP 2% (paraphormaldehyde-lysine-periodate sodium). Three sections (20 μm thick) were randomly obtained from each footpad and immunostained with rabbit polyclonal antiprotein gene product 9.5 (UCHL1/PGP 9.5 (protein gene product 9.5) Rabbit Polyclonal antibody, Proteintech, Illinois, Rosemont, IL, USA) using a free-floating protocol. The total number of PGP-positive fibers IENFs were counted under a light microscope (Nexcope NE 920 light microscopel TIEsseLab S.r.L., Milan, Italy) at 40X magnification with the assistance of a microscope-mounted video camera. Individual fibers were counted (blind examiner) as they crossed the dermal–epidermal junction, and secondary branching within the epidermis will be excluded. A computerised system measured epidermidis length to calculate the linear density of IENF/mm.

#### 4.2.7. Immunohistochemistry for NCX2

DRG of three animals per group were dissected, fixed in 10% formalin for 3 h at RT and paraffin embedded. Three-μm-thick slices were cut with a Leica RM2265 microtome (Microsystems GmbH, Wetzlar, Germany). Immunohistochemistry was performed using a rabbit polyclonal anti-Na^+^/Ca^2+^ Exchanger 2 antibody (NCX-2, TA328916, OriGene, Rockville, MD, USA). Paraffin sections were deparaffinised with xylene, rehydrated, and heated in a steamer for 20 min (1 mM EDTA pH 7.4) to retrieve antigens. Endogenous peroxidase activity was quenched by incubation in 3% H_2_O_2_ for 10 min at RT. The slides were washed in PBS and incubated in 5% NGS for 1 h at RT. Then, the sections were incubated with anti-NCX-2 antibody (1:200 in 1% NGS) o/n at 4 °C. The following day, the slides were washed and incubated with a biotinylated secondary antibody to rabbit IgG for 1 h at RT (1:100, Vector Laboratories, Peterborough, UK) followed by incubation with streptavidin-conjugated horseradish peroxidase for 1 h at RT (1:100, ABC kit Vectastain, Vector Laboratories, Peterborough, UK). The antigen–antibody complex was visualised by incubating the sections with 3,3-diaminobenzidine hydrochloride (DAB) (Sigma, St. Louis, MO, USA) dissolved in PBS with 10 μL of 3% H_2_O_2_. Negative controls were incubated only with the secondary antibody. DAB intensity of images was then quantified using ImageJ software and Colour Deconvolution plugin.

#### 4.2.8. Immunofluorescence for NCX2

Three-μm-thick paraffin DRG sections were deparaffinised with xylene and rehydrated and non-specific tissue binding was blocked with 5% NGS and 5% BSA in PBS for 1 h. Samples were then incubated overnight at 4 ℃ with NCX2 primary rabbit antibody (1:200, OriGene, Rockville, MD, USA) and with a goat anti-rabbit secondary antibody Alexa red 546 (1:200, Invitrogen Waltham, MA, USA) for 1 h at room temperature. Sections were then examined, and stitched images were acquired using an epifluorescence microscope (Cell observer. Zeiss, Germany). Fluorescence intensity of images was then quantified using ImageJ software.

#### 4.2.9. Western Blotting for NCX2

DRG specimens were harvested at sacrifice at end of treatment, and proteins were extracted after chemical and mechanical lysis. Lysis solution was made up of 10% glycerol, 25 mM TrisHCl pH 7.5, 1% Triton X-100, 5 mM EDTA pH 8.0, 1 mM EGTA Ph 8.0, 10 mM sodium orthovanadate, 4 mM PMSF, 1% aprotinin, and 20 mM sodium pyrophosphate. Protein concentration was quantified by Bradford method, and 10 µg were denatured and loaded onto 10% SDS-PAGE. After electrophoresis, proteins were transferred to nitrocellulose membranes and immunoblotting analysis was performed following manufacturer instructions. Briefly, membranes were blocked with 5% non-fat milk blocking solution and then incubated with primary antibodies against NCX2 (1:500, OriGene, Rockville, MD, USA) and beta actin (1:1000, Santa Cruz Biotechnology, Dallas, TX, USA). After incubation, membranes were washed and then incubated with appropriate horseradish peroxidase-conjugated secondary antibodies (anti-rabbit, 1:1000, PerkinElmer, Waltham, MA, USA; anti-goat, 1:2000 Santa Cruz Biotechnology, Dallas, TX, USA). Immunoreactive proteins were visualised using an ECL chemiluminescence system (Amersham, Sullivan County, TE, USA).

#### 4.2.10. NCS

Nerve conduction studies were performed with the electromyography apparatus Matrix Light (Micromed, Mogliano Veneto, Italy) and performed as previously described [72]. Briefly, subdermal needle electrodes were used (Ambu Neuroline (Ambu, Ballerup, Denmark)). Recordings were performed under deep isofluorane anesthesia and animal body temperature was monitored and kept constant at 37 ± 0.5 °C with a thermal pad (Harvard Apparatus, Holliston, MA, USA). Caudal nerve and digital nerves were assessed. Caudal nerve conduction study was performed positioning a pair of recording needle electrodes at the base of the tail (inter-electrode distance: 0.5 cm) and a pair of stimulating needle electrodes (inter-electrode distance: 0.5 cm) was placed 3.5 cm distally with respect to the active recording electrode; the ground electrode was placed 1 cm distally to the active recording electrode. Digital nerve conduction study was obtained positioning the positive recording electrode in front of the patellar bone, the negative recording electrode close to ankle bone, the positive and negative stimulating electrodes close to the fourth toe near the digital nerve and under the paw, respectively; the ground electrode was placed in the sole. Intensity, duration, and frequency of stimulation were set up in order to obtain optimal results. Averaging technique was applied carefully. Recordings filters were kept between 20 Hz and 3 KHz. Sweep was kept at 0.5 ms.

#### 4.2.11. NET

During recordings, mice were under deep isoflurane anaesthesia and body temperature was kept constant at 37 ± 0.5 °C, as stated above. The montage to perform caudal nerve recordings was adapted slightly, modifying the previously devised protocol by Boërio et al. [73]. Briefly, disposable, non-polarizable, subdermal platinum iridium needle electrodes (MedCat Supplies, Klazienaveen, The Netherlands) were used for stimulation; as ground and recording electrodes, subdermal needle electrodes were instead used (Ambu Neuroline (Ambu, Ballerup, Denmark)). The stimulating cathode was placed at the base of the tail and the anode was placed in the base of the rear foot, on the same side of the caudal nerve tested (left). Recording needle electrodes were inserted subcutaneously in the left side of the tail (interelectrode distance: 1 cm), 3.5 cm distally respect to the cathode. Ground electrode was inserted in the right side of the tail, in mid between the cathode and the active recording electrode. As a stimulator, an isolated linear bipolar constant current stimulant (maximal output ±10 mA, DS4, Digitimer, Welwyn Garden City, UK); the Xcell3 Microelectrode Amplifier (FHC, Bowdoin, ME) was connected to the recording electrodes via a customised probe/adapter specifically designed by FHC for our needs; National Instrument USB-6221-BNC Acquisition Device (National Instrument Italy, Assago, Italy) was used to connect all these instruments. For NET recordings, Qtrac© software (Institute of Neurology, Queen Square, London, UK) and TROND protocol were used. For the purposes of this study, Recovery Cycle is the curve of interest. Recovery Cycle was performed using paired pulse stimuli: a supramaximal stimulus followed by a conditioning one at different interstimulus intervals (2–200 ms) [74]. Threshold changes at 2 and 2.5 ms interstimulus intervals were used to determine refractoriness. The minimum mean threshold change of three adjacent points defined superexcitability; instead, subexcitability was determined as the minimum mean threshold change obtained after a 10 ms interstimulus intervals.

#### 4.2.12. Statistical Analyses

Descriptive statistics were generated for all variables. Normally distributed data were analyzed with parametric tests (t-test, one-way Anova followed by Dunnet test) and non-normally distributed with non-parametric tests (Mann–Whitney U-test, Kruskal–Wallis test, followed by pairwise comparison, adjusted by the Bonferrroni correction for multiple tests). Two-sided tests were used. A *p*-value < 0.05 was set as significant. All analyses were conducted in GraphPad environment (v4.0), apart from NET recordings. The latter were analyzed with QtraS© (Institute of Neurology, Queen Square, London, UK), specifically designed to dialogue with the recording software QtraC© (Institute of Neurology, Queen Square, London, UK).

## 5. Conclusions

Our journey from the bench side to bed side and vice versa allowed us to pave the way to a new line of research exploiting NCX2 modulation to prevent OIPN-related ADa, highlighting a novel druggable target to cure this condition and, potentially, other conditions in which ADa ensues, as consequences of Na^+^VOC/NCX family alterations.

## Figures and Tables

**Figure 1 ijms-23-10063-f001:**
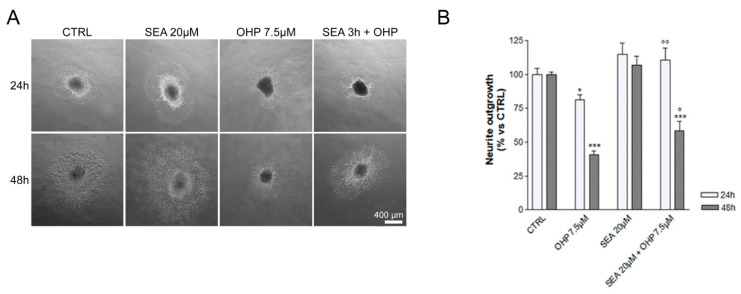
In vitro experiment results. (**A**) shows representative images of neurite elongation at 24 and 48 h (upper and lower panel, respectively); CTRL: control group; SEA: SEA0400-treated; OHP: Oxaliplatin-treated. (**B**) shows neurite outgrowth as a percentage respect to the CTRL group (statistical significance of one-way ANOVA (followed by Tukey’s Multiple Comparison Test) is also provided: * *p* < 0.05 vs. CTRL; *** *p* < 0.001 vs. CTRL; ° *p* < 0.05 vs. OHP, °° *p* < 0.01 vs. OHP).

**Figure 2 ijms-23-10063-f002:**
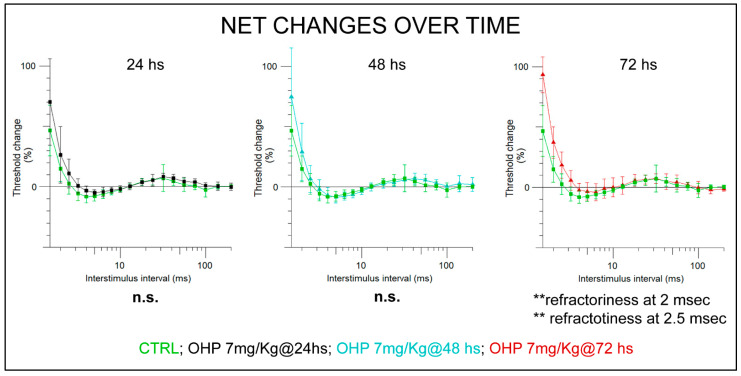
NET findings at recovery cycle at 24, 48, 72 h following 1st Oxaliplatin i.v., administration. Statistical significance of Mann–Whitney U-test is shown. In the graph, individual values are accompanied by SEM bars (** *p*-value < 0.01 vs. CTRL). CTRL: control group (green curves); OHP: OHP-treated group (curves at 24 h are shown in black, at 48 h in cyan, and at 72 h in red).

**Figure 3 ijms-23-10063-f003:**
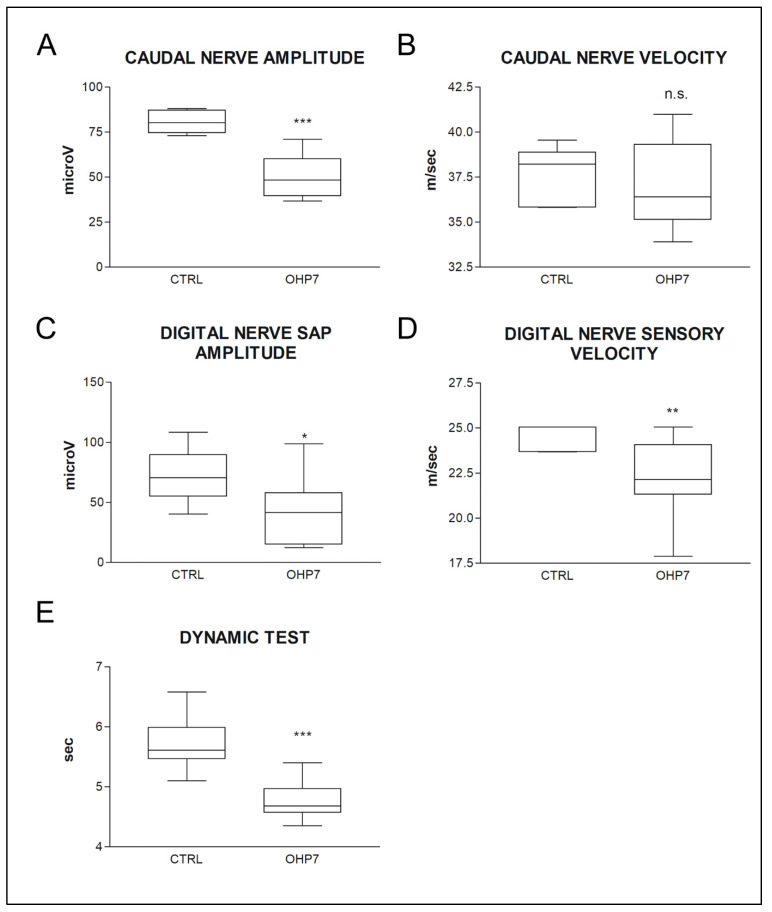
In (**A**–**D**): nerve conduction study data at the end of treatment is shown. In (**E**) Dynamic (mechanical allodynia) test data at the end of treatment is shown. Statistical significance of Mann–Whitney U-test is shown in all graphs. The box-and-whiskers graphs show median and quartile values, as well as maximum and minimum values. * *p*-value < 0.05 vs. CTRL, ** *p*-value < 0.01 vs. CTRL, *** *p*-value < 0.001 vs. CTRL. CTRL: control group; OHP7: OHP-treated group. SAP: sensory action potential. n.s. means no significance.

**Figure 4 ijms-23-10063-f004:**
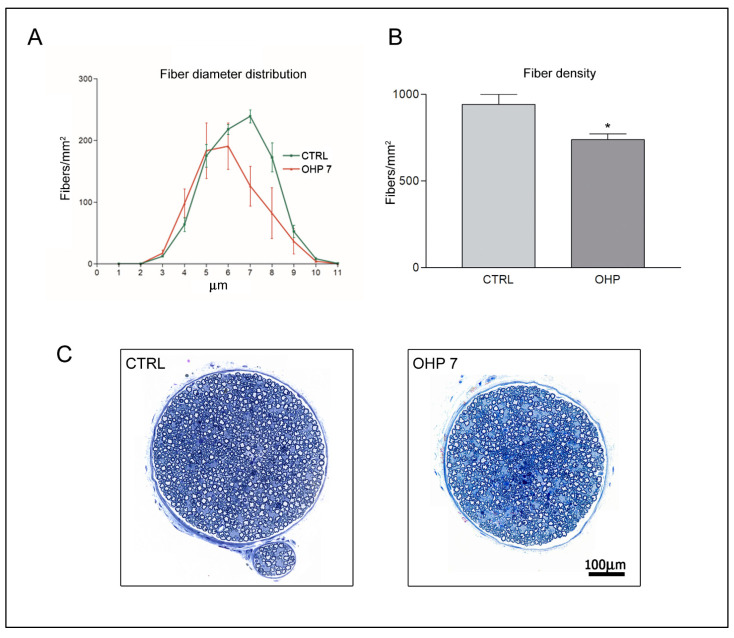
Morphological and morphometrical assessment of caudal nerves at the end of treatment. (**A**) shows the graph of the distribution of fiber diameters (SEM bars are represented). (**B**) shows the box-plot graph (standard deviation bar is shown) of statistical analysis (*t*-test) of morphometry. In (**C**)) representative images of CTRL and OHP animals are shown, to highlight the mild axonal loss in caudal nerves of the OHP group: fiber density is moderately diminished and degenerating fibers are visible in OHP group. CTRL: control group; OHP7: OHP-treated group. * *p*-value < 0.05 vs. CTRL.

**Figure 5 ijms-23-10063-f005:**
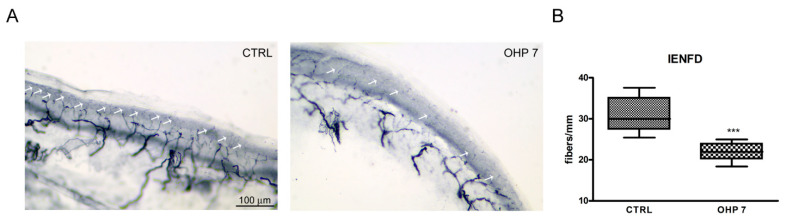
IENFD at the end of treatment. In (**A**), representative images of IEFND in both groups are shown; white arrows point out small fibers visible in each photograph. In (**B**), the graph representing the statistical significance of Mann–Whitney U-test is shown (in the box-and-whiskers graphs median and quartile values, as well as maximum and minimum values, are shown; *** *p* < 0.001 vs. CTRL). CTRL: control group; OHP7: OHP-treated group.

**Figure 6 ijms-23-10063-f006:**
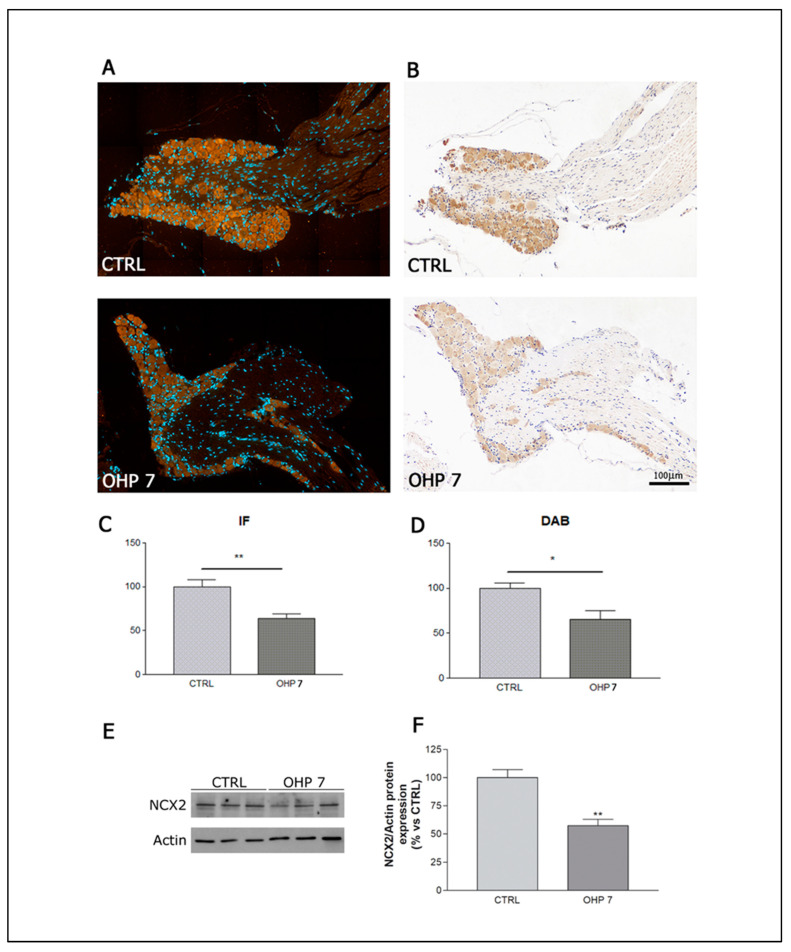
Immunohistochemistry, immunofluorescence, and western blotting for NCX2. On the upper panels representative images of immunofluorescence (**A**) and immunohistochemistry (**B**) are shown, accompanied by graphs showing the statistical significance (*t*-test) for IF and DAB quantification, in (**C**,**D**), respectively). In the bottom panel, the running lane of the western blotting (**E**) is shown as well as the graph (**F**), showing the statistical significance in NCX2 quantification (*t*-test). CTRL: control group; DAB: 3, 3’-diaminobenzidine staining in immunohistochemistry; IF: immunofluorescence; OHP7: OHP-treated group. For each column graph, the standard deviation bar is represented. * *p*-value < 0.05 vs. CTRL; ** *p*-value < 0.001 vs. CTRL.

## Data Availability

Data will be made available upon request to the corresponding author.

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
