# Peer review of "Sodium-Calcium Exchanger 2: A Pivotal Role in Oxaliplatin Induced Peripheral Neurotoxicity and Axonal Damage?"

_ijms, 2022, doi:10.3390/ijms231710063_

Round 1

Reviewer 1 Report

The authors propose to investigate the role of NCX2 in oxaliplatin (OHP) induced peripheral neurotoxicity (OIPN) which results in axonal damage.  They do show that a blocker of NCX2 prevents decreased DRG explant neurite elongation induced by OHP (Figure 1) and that OHP induces changes in NCX2 expression profile (Figure 6) in DRG harvested from animals after 8 weeks OHP.  The rest of the data just establish the damaging effects of OHP treatment.  As the authors themselves state in the discussion, these results indicate a potential involvement of NCX2 in OIPN but no real evidence of the proposed mechanism (reversal of transporter) is shown.

In general the data are not introduced sufficiently.  For each figure, the author should explain what the test is, why they chose to perform it, and what the results mean.  There are many results of tests presented that, to anyone not in the specialized field, are not interpretable  (e.g. "dynamic test", IENFD..). Also what subject of the experiment  should be expressly presented  (eg explants were from rats, others involve mice) without having to look it up in the methods.

The figures also mainly present averaged data where it is customary to first show individual traces to help understand the test and the quality of the data.

Author Response

Reply to Reviwer1’s comments.

The authors propose to investigate the role of NCX2 in oxaliplatin (OHP) induced peripheral neurotoxicity (OIPN) which results in axonal damage.  They do show that a blocker of NCX2 prevents decreased DRG explant neurite elongation induced by OHP (Figure 1) and that OHP induces changes in NCX2 expression profile (Figure 6) in DRG harvested from animals after 8 weeks OHP.  The rest of the data just establish the damaging effects of OHP treatment.  As the authors themselves state in the discussion, these results indicate a potential involvement of NCX2 in OIPN but no real evidence of the proposed mechanism (reversal of transporter) is shown.

R: We are thankful for this comment since it enables us to better elucidate our reasoning. Our data are showing, even if indirectly, that reverse mode activation took place. First of all, we demonstrated that in the in vivo model, the prerequisite of reverse mode activation ensued (i.e., acute sodium voltage operated ion channel dysfunction) via nerve excitability testing as soon as after the 1st oxaliplatin injection, since the recovery cycle curve showed a significant upward shift in the oxaliplatin group; this is the trigger to switch the mode of NCX functioning. If reverse mode is activated via an aberrant depolarisation (as oxaliplatin is able to generate) neurons downregulate NCX to avoid a calcium overload (which would be the final effect of a prolonged reverse mode activation). To support this statement we already reported the paper published by Boscia et al.: in line 269-271 we stated, in fact, “in in vivo models of damage determining repetitive spreading of depolarization current (i.e., as occurring during brain ischemia), a downregulation of NCX2 was observed in neurons”. We are thankful for this comment since we were not able to fully explain this point: we have highlighted the concept more extensively in the current revised version of the text: we revised result (line 199-210) and discussion section (line 292-303). In our in vivo model we demonstrated, quite robustly (via western blotting, immunohistochemistry and immunofluorescence) and with a statistical significance, that NCX2 was downregulated in oxaliplatin animals. Therefore, the 2 months of treatment and repetitive oxaliplatin administration (each injection is a trigger for reverse mode activation) were able to determine an enough relevant reverse mode activation that the downregulation mechanism was established; this is an endogenous and autoprotective mechanism that it is not ultimately strong enough to prevent axonal damage though. In line with this reasoning (i.e., that reverse mode activation was indeed present and also relevant in causing axonal damage), in vitro data shows that blocking the NCX axis via SEA0400 axonal damage is modulated.

In general the data are not introduced sufficiently.  For each figure, the author should explain what the test is, why they chose to perform it, and what the results mean.  There are many results of tests presented that, to anyone not in the specialized field, are not interpretable  (e.g. "dynamic test", IENFD..). Also what subject of the experiment  should be expressly presented (eg explants were from rats, others involve mice) without having to look it up in the methods.

R: The style of the journal requires that methods are described at the end of the paper itself. We are thankful for this comment. To enhance readability we revised the result section (changes are shown in revision mode) to better explain the relevance and meaning of presented data without altering the journal style.

The figures also mainly present averaged data where it is customary to first show individual traces to help understand the test and the quality of the data.

R: only Figure 1 shows data with only mean values, while others represent all relevant descriptive statistics (we better described this in each figure caption). We therefore revised the Figure 1 showing mean +/-SEM values.

For what regards traces (i.e., neurophysiological recordings):

  • Nerve conduction studies (Figure 3) are not conventionally shown as a superimpose of the single trace of each individual recording but with graphs/tables addressing median/quartiles/min and max values (in case, as we did, non-parametric statistics were performed) or mean values+/-standard deviation (if data are normally distributed and analysed with parametric tests), to prove the appropriateness of data. As a reference of this convention we report our previously clinical and preclinical papers using nerve conduction studies in OIPN setting: PMID: 29594485, PMID: 29594485, PMID: 31811874, PMID: 33499072, PMID: 34391792. Therefore, we did not update Figure 3 but better described in the Figure caption data represented in the box-and-whiskers graphs.
  • Nerve excitability testing can greatly benefit from a superimpose graph to show single traces, therefore, we are most thankful for the comment and we ameliorated provided figures. However, the superimpose mode makes it quite difficult to appreciate the trend of the group. Therefore, instead of using the superimpose mode in the already revised Figure 1 as stated above, we provided a supplementary Figure1 that shows single traces for each single group to allow a better readability of the image and reported data.

Author Response

Reply to Reviwer2’s comments.

In this study, the investigators intended to study if NCX2 inhibitors (e.g.,SEA0400) can ameliorate oxaliplatin-induced neuropathy related to axonal damage. There are several queries regarding the present investigations, which are shown in the following.

Major comments:

[1] SEA0400 itself might be not specific simply for being an inhibitor of reverse Na+ /Ca2+ exchanger, although it appears to be a potent inhibitor. For example, is it possible that SEA0400 is also likely to exert additional off-target inhibitory effect on variable types of plasmalemmal ionic currents, e.g., voltage-gated Na+ current or funny current in sensory neurons? Alternatively, the reduction in late Na+ current by small molecules (e.g., ranolazine) can result in a diminution in Ca2+ overload by increasing the driving force for Ca2+ extrusion through the NCX process that is functioning in reverse mode (i.e., in a mechanism that operates to extrude Ca2+ in exchange for the influx of Na+ ) (refs: Soliman et al., J Pharmacol Exp Ther 2012;343:325-332; Shenoda, Transl Stroke Res 2015;6:181-190). Please comment on these queries on the Discussion section of the manuscript, since this issue is of relevance in this manuscript.

R: we are thankful for this comment. Previously published works on neurons of the peripheral and not the central nervous system (27166151) showed that SEA0400 efficacy was obtained using microM concentrations and similar observations were performed on Sh-SY5Y cell cultures (21672583). The specific reasons for the drug selection and the scaling dosages are further discussed in reply to major comments [2] and minor comment [8]. Therefore, we do not expect that relevant off-targets are implicated. However, as we better discuss in the discussion and conclusion we propose a proof-of-concept and pilot study. The in vivo data clearly points out that NCX involvement is possible. The in vitro data we present aim at paving the way to a new line of research. Thus, we do not suggest that SE0400 is the drug to be used against OIPN, but that our data support the appropriateness of further investigation exploiting selective approaches, such as verifying effects on NCX2 modulation exploiting innovative technologies such as siRNA.

[2] In the Discussion section, some of the texts appear to be irrelevant to the results shown in the manuscript and hence need to be appropriately revised and modified. For example, the statement which focused on NCX activity and consequent actions appears to be over-emphasized to some extent. If that would be possible, several different inhibitors of NCX exchangers other than SEA0400 need to be further and comparatively evaluated.

R: we are thankful for this comment. Our experiments aim at providing a proof-of-concept and pilot study to verify if modulating the NCX axis some benefits are to be expected. Among possible inhibitors, we selected SEA0400 since it is the most potent and selective inhibitor of NCX (PMID: 11408549) and it was already studied in the nervous system in brain ischemia, a setting in which a) first sodium currents are altered and b) then NCX reverse mode is activated. The less strong NCX inhibitor KB-R7943 showed a rather low specificity for NCX family only even at low dosages with larger impact on other transporters than SE0400 (PMID: 11118291, 11408549, 15592578). Therefore, SE0400 appeared a fair option to test our hypothesis; moreover, SEA0400 were used in these studies that already investigated nervous system injury due to Ca2+ paradox (i.e., Ca2+ toxicity due to NCX2 reverse mode activation secondary to an alterations of sodium currents): Matsuda et al. 2001 (PMID: 11408549) showed that SEA0400 attenuated dose-dependently paradoxical Ca2+ challenge-induced production of reactive oxygen species, DNA ladder formation, and nuclear condensation in cultured astrocytes. Koyama et al. 2004 (PMID: 15087242) similarly showed that this compound was again efficacious in brain ischemia. Moreover, in a model of peripheral neuropathy due to paclitaxel and not oxaliplatin, it was shown that some neuronal populations of the peripheral nervous system can be resistant to the effect of KB-R7943, but responsive to SEA0400 (27166151). Therefore, we selected this drug as being the most appropriate for the specific system we were investigating and the specific pathogenetic hypothesis (the presence of an altered sodium current). We revised extensively the Introduction and Discussion section to take into account this comment (in the introduction is presented the general literature on NCX and its functioning to avoid irrelevant information in the discussion).

[3] Oxaliplatin (OHP) has been previously reported to affect the magnitude and gating kinetics in different types of transmembrane ionic currents inherently present in excitable cells, e g., voltage-gated Na+ current and hyperpolarization-activated cationic current (ref: Wu et al., 2009; Neurotoxicology 2009;30(4):677-685). Please comment on this issue as well. Is it thus likely that SEA0400 itself affects these types of ionic currents directly? Please also quote this paper (Morris et al., Left-shifted Nav channels in injured bilayer: primary targets for neuroprotective Nav antagonists? Front Pharmacol 2012;3:19), since it is relevant to the present results about nerve injuries produced by OHP (i.e., OIPN).

R: thank you for this comment. In our hypothesis sodium currents are the main target of oxaliplatin neurotoxicity (we better elucidate starting from line 99 in the introduction where we quote papers suggested by Reviewer2; line 85-87 and line 99-104). For what regards the paper by Morris et al. suggested by Rev.2, however, an important anatomical/pathological note should be given: in this review a specific attention is given to “sick NaV channels” but the specific OHP-related damage is different from the ones described in this review. Alterations of sodium currents are transient and no structural alterations of plasmalemma are present (32519566). Therefore, NCX is even more an interesting target: NaV do not encounter persistent alterations and some downstream events can thus be hypothesised. As stated above, we did not expect that relevant off-targets to be implicated, but we better discussed this (see sections modified in reply to major comment [1] and line 363-365 in particular), recognising the limitations of our study and the necessity of further studies (our is a proof of concept and pilot study).

[4] In the Introduction section, the paragraph appears to be lengthy and needs to be appropriately separated into two parts. The rationale of the present study needs to be emphasized more clearly. For instance, why was NCX inhibitor (e.g., SEA0400) chosen in the present study. For example, in line 94, “NCX is ……” was changed and moved down to the second paragraph in the Introduction section of the manuscript.

R: thank you for the thoughtful suggestion. We revised the Introduction and Discussion as already described in reply to major comment [2].

Minor comments:

[1] In line 68, please describe the definition of “double-faced syndrome” in detail.

R: we modified the sentence to better explain the concept. Thank you.

[2] In line 117, SEAA00400 needs to be replaced with SEA0400.

R: modified as suggested. Thank you.

[3] In line 306, Dimethyl Sulfoxide needs to be replaced with “dimethyl sulfoxide”. It is unnecessary to be capitalized.

R: modified as suggested. Thank you.

[4] In line 317, Calf Bovine Serum is not needed to be capitalized.

R: modified as suggested. Thank you.

[5] In line 319, CO2 should be changed to CO2 (i.e., in subscript).

R: modified as suggested. Thank you.

[5] The superscripts in Na+ and Ca2+ appearing throughout the entire text of the manuscript are necessarily corrected.

R: modified as suggested. Thank you.

[6] In Figures 3, 5 and 6, please indicate the meaning of *, ** and ** shown in the relevant panel as demonstrated in the legend of Figure 1.

R: we corrected captions. Thank you.

[7] In Figure 4A, the resolution needs to be improved (e.g., the thickness of error bar).

R: we modified the images as suggested. Thank you.

[8] Are there any specific reason(s) why the investigators involved chose the OHP concentration at “7.5 mM” or SEA0400 at “20 mM”? Since SEA0400 is thought to be very potent in the inhibition of NCX process particularly at the reverse mode (line 279 in the manuscript), the concentration (i.e., 20 mM) used in the study could be noticeably quite higher, and the other unidentified actions by SEA0400 might hence occur and be underestimated. Please comment on this issue in the Discussion section of the revised manuscript.

R: the specific model of disease and system we are dealing with has some specific characteristics to be taken into account. Therefore, experiments stated above concerning the central nervous system can be a good starting point for the drug selection (mostly nM concentrations were used), but we considered other published works on the squid giant axon which demonstrated that to have actual effects on axons a higher – despite still selective – dosage is required, in the range of similar microM to ours (PMID: 15592578), that better resembles the peripheral nervous system setting our disease model. Moreover, as stated above, in a different model of peripheral neuropathy SEA0400 was used in similar microM concentrations, as well as in neuroprotection experiments relying on SH-SY5Y cells. On the basis of this, we tested scaling dosages of 1, 10, 20 microM.

For what regards selection of oxaliplatin dose, we relied on the extensive review of Calls et al. (PMID: 31865195) which presented an overview of the in vitro and in vivo models of OIPN. We selected a dosage that was in the range of the established in vitro dosages that can determine alterations in sodium currents, on the basis of plasma mean Cmax in patients treated with Oxaliplatin (PMID: 10778943, 12453338); since DRG are characterised by fenestrated capillaries, this is the dosage that they are exposed to in the clinical setting. Since dosages lower than the one we selected (7.5microM) were not able to induce significant alterations in neurite elongation (unpublished data of our lab) and, therefore, this was the best compromise to recreate a model that was reproducing in vitro both sodium current alterations and axonal damage.

Round 2

Reviewer 1 Report

The authors have adequately addressed my previous comments.

Reviewer 2 Report

acceptable !